# Online therapy with families - what can families tell us about how to do this well? A qualitative study assessing families' experience of remote Dyadic Developmental Psychotherapy compared to face-to-face therapy

**Monica Blair**[1]☯*, **Leigh Tweedlie**[1]☯, **Helen Minnis**[2], **Irene Cronin**[3], **Fiona Turner**[2]

**1** College of Medical Veterinary and Life Sciences, University of Glasgow, Glasgow, United Kingdom, **2** Institute of Health & Wellbeing, University of Glasgow, Glasgow, United Kingdom, **3** Academic Child and Mental Health Services, University of Glasgow, Glasgow, United Kingdom

☯ These authors contributed equally to this work.

* 2364112b@student.gla.ac.uk

**Data Availability Statement:** All relevant data are within the manuscript and its Supporting Information files.

## Abstract

Dyadic Developmental Psychotherapy (DDP) is a family-based therapy for adopted children aiming to achieve secure attachment between the child and parent. Due to restrictions under the COVID-19 pandemic, delivery of DDP transitioned from face-to-face to online methods. This study aimed to explore families experience of online DDP compared to face-to-face DDP, looking at the advantages and disadvantages of remote delivery methods and the implications this has on future service delivery for clinicians. Semi-structured interviews with 6 families were conducted online. Analysis of transcripts using Interpretative Phenomenological Analysis (IPA) revealed four superordinate themes: *environment and child engagement*, *non-verbal communication*, *travel* and *familiarity with remote interactions*. Parents recognised the influence the physical and online environment had on their child's engagement levels, however, varied in their experience and hence preference of delivery method. All families emphasised the importance of non-verbal communication within DDP sessions and majority highlighted this may be lost online. For families who travelled to face-to-face DDP, car journeys provided a unique opportunity to decompress and reflect after sessions. For families where travel is unfeasible, online DDP was a lifeline, demonstrating the ability of remote therapy to widen access to specialist healthcare. Familiarity with online work emerged as a strong indicator of positive attitudes towards remote DDP, especially if the previous experience is positive and the child is confident using technology. Overall, families differed greatly in their experience of remote and face-to-face DDP indicating a new approach must be undertaken with each family beginning therapy, ensuring it is unique and individual to their needs.

**Funding:** HM NIHR 127801 National Institute for Health and Care Research https://www.nihr.ac.uk The funders had no role in study design, data collection and analysis, decision to publish, or preparation of the manuscript.

**Competing interests:** The authors have declared that no competing interests exist.

## Introduction

The outbreak of COVID-19 created a tipping point for remote work within mental health services, triggering a mass migration of various interventions to online delivery, including family therapy [1]. For families themselves, the pandemic imposed a tidal wave of new challenges; disruption to routines, altering childcare, school closures, dissolving extra-curricular activities as well as bringing financial hardship and job uncertainty [2]. Furthermore, the restrictions and changes enforced on families, coupled with a rise in parental stress, has heightened the risk of child abuse and maltreatment, which worryingly may have gone underreported [3–5]. The combination of these factors emphasised the importance of specialist therapeutic care to continue during the pandemic remotely, especially for families with children who have a history of maltreatment.

Online (often called remote) therapy, defined under the umbrella term "tele-mental health", involves mental health services delivered via telephone or video appointment. Evidence for the efficacy of remote family therapy has been gathered and assessed, with multiple studies reporting statistically significant improvements in all mental health variables measured [6]. Meta-analysis of these studies concluded that online therapy is equivalent to face-to-face therapy in terms of child behaviour outcomes and parental depression with further analysis demonstrating remote therapy as superior to treatment as usual, written resources, or waiting list controls [7]. Improved service access, availability and flexibility is a primary advantage of remote delivery methods and is commonly recognised across various psychotherapies, including family therapy [8, 9]. Disparity in access to specialist psychotherapy within rural, low-income, and ethnic minority communities was highlighted during COVID-19 [10]. Telemental-health may therefore provide a path to specialist healthcare for disadvantaged families and reduce the burden of health inequality caused by lack of resources for those living in rural areas of the UK. On the other hand, prominent ethical concerns have emerged regarding the negative impact of remote therapy on factors such as confidentiality, risk, and interpreting non-verbal cues [8, 11]. Ensuring confidentiality throughout a remote therapy session or encouraging a flowing conversation with slow internet connections adds a new layer of complexity to the role of the therapist. Using unsecured websites and free to use software could lead to compromised data security [9]. On an individual level, it can be difficult for patients to find a space that is entirely private in their home which could hinder their willingness to be open and honest with the therapist. Professional regulatory bodies in the UK have begun to publish ethical guidelines for online therapy sessions, which all practitioners registered with these bodies must adhere to [12, 13]. However, there is still a lack of legislation for online practice on a regional and national level, and no adequate governance of online therapies across national borders exists.

In conjunction with the growing number of studies into the efficacy of remote family therapy, an evidence-base of qualitative research has emerged since the pandemic, focusing on the experience of family therapists. Cronin et al. shed light, in a qualitative study of therapist views, on how the environment surrounding online therapy influences many aspects of the therapeutic encounter [14]. Furthermore, the researchers posited that increased participant confidence was related to positive attitudes towards online family therapy. Survey results from UK family therapists working remotely during COVID-19 restrictions concluded that the majority reported an overall positive experience of online therapy; citing continuity of work, effective use of time and improved accessibility as factors contributing to this [15]. The main negatives identified were digital exclusion (lack of access to, or the appropriate skills to utilise digital devices such as a smartphone or a computer), assessment and management of risk

concerns, as well as difficulties engaging people and identifying non-verbal cues. Challenges unique to family therapy settings have also transpired such as working with multiple people, navigating distractions in the home environment and managing challenging situations such as tantrums during online therapy sessions [16]. In light of this, researchers suggest that adaptations need to be made when delivering interventions via online platforms with families such as special considerations regarding the environment and privacy. Despite this informative research, accounts from families themselves have largely been neglected. Every family has their own distinctive experience and perspective of therapy which can give context to clinical evidence and therefore must be included in current research [17].

Remote and rural areas are not well served by specialist therapies, however, online delivery methods could widen access for families who live outside of urban areas. A scoping review of mental health services in remote areas of high-income countries reported that delivering therapy to those who live in rural areas has several positive outcomes [18]. These include lessening the financial burden of therapy and increased confidence in technology. Providing specialist therapy in a remote setting may improve access to healthcare and positively impact rural communities. Dyadic Developmental Psychotherapy (DDP) is one specialist family-based therapy which transitioned to remote delivery during the pandemic. DDP was developed as a treatment for adoptive families with children who have a history of abuse and have not responded to traditional psychological therapies such as Cognitive Behavioural Therapy [19]. Due to the nature of their difficulties, these children often have complex issues surrounding trust and struggle to reflect and regulate emotions when exploring their experiences within those therapy models [20]. Rooted in attachment theory; the primary goal of DDP is to facilitate the developmental qualities associated with attachment security between the child and parents [21]. These qualities include the capacity to trust, regulate, and reflect about oneself and others. The therapeutic and parenting attitude of DDP is encapsulated by the acronym PACE; *playfulness*, *acceptance*, *curiosity* and *empathy*. Using PACE, the therapist helps the child to regulate their emotions as they explore scenarios that could potentially trigger traumatic memories, whilst also facilitating emotional regulation between the child and caregiver. Therapists and parents will use their intersubjective view of the child to help them co-construct alternative narratives of their past experiences. Through multiple DDP sessions, the child's negative conclusions about themselves because of their experiences are revised, leaving them with a greater sense of trust, less shame-based behaviours, and more regulation within their lives.

DDP sessions are tailored to each family and their needs, lasting between one to two hours per session, with the total number of sessions ranging depending on the family [19]. Initial DDP sessions are parent-only, to foster a strong therapeutic alliance and allow the parents to better understand their role in the therapeutic process. The therapist will encourage conversations around attachment styles in the caregiver's own up-brining to identify what elements of parenting are particularly challenging for them. By developing a deeper understanding of their unique triggers, caregivers can in turn build a greater sense of empathy for their child during challenging scenarios. The child will then join the sessions when the therapist and parents are ready.

Adopted and fostered children who have experienced early trauma have higher rates of attachment disorders such as Reactive Attachment Disorder (RAD) and Disinhibited social engagement disorder (DSED) [22, 23]. Emotional dysregulation and behavioural difficulties are common in children with attachment disorders and neurodevelopmental conditions such as Attention Deficit Disorder and Autism Spectrum Disorder are also more prevalent in children who have experienced abuse and neglect [24–26]. Publications from Loughborough University and the National Society for the Prevention of Cruelty to Children (NSPCC) have stated that it is more expensive to allow a child in care to be unsupported and unstable, than to provide them with specialist, nuanced mental health care [27, 28]. Development and

evaluation of specialist quality interventions for adoptive children with experiences of mal-treatment, such as DDP, are therefore not only important for them as individuals, but also for wider public health.

The National Institute for Health and Care Excellence has recognised and reacted to this need; recommending quality evidence-based research be conducted into DDP National Institute for Health and Care Excellence (NICE) [29], leading to the Relationships in Good Hands Trial (RIGHT). RIGHT is an ongoing UK-based randomised control trial that aims to assess the clinical and cost-effectiveness of DDP compared to services as usual [30, 31]. With the exception of one investigation into parents' experiences of face-to-face DDP and one exploration of adoptive parents' experience of a support group based on DDP principles [32, 33], like many family therapies, the perspectives of families have widely gone unheard and no study has yielded any information on remote delivery methods of DDP. In order to fully evaluate a complex intervention like DDP, the quantitative RCT research should be consolidated with quality qualitative research methods and process evaluation [34].

The overall aim of this study is to explore families' lived experiences of remote DDP compared to their experiences of face-to-face DDP, with a specific focus on advantages and disadvantages of remote DDP delivery. The insights gained from these families will be used to contribute to recommendations on how to improve the experience of remote interactions and inform a more equitable delivery of DDP.

## Methods

### Research design

Qualitative semi-structured interviews were conducted with parents or carers from 6 families and analysed using interpretative phenomenological analysis (IPA). IPA allows researchers to gain insight into participants' personal lived experience and the meaning-making that occurs in relation to their experiences [35, 36]. The phenomenon investigated was the family's personal experience of DDP from the parent perspective. In IPA, interpretation of meaning is multi-layered as the researchers make sense of participants' meaning-making of the phenomenon, known as 'double hermeneutic' [37]. DDP is intrinsically a deeply personal experience given the nature of the problems it strives to explore and resolve. Additionally, each child and their adoptive parents will have had unique and individual experiences that has brought them to the attention of needing an intervention like DDP. IPA's central focus on individual experience therefore makes it a suitable methodology to explore DDP experience. The consolidated criteria for reporting qualitative research (COREQ) was used to aid reporting of the study design [38]. Ethical approval was obtained by West of Scotland Research Ethics Committee 3. All participants gave written informed consent to be involved in the study.

### Participants

Participant demographics can be found in Table 1. The names of the parents, children, and therapists have been changed to protect anonymity. Six qualitative semi-structured interviews were conducted with six parents and one long-term foster carer, from six families, who have a child that suffered maltreatment within their birth family and are now receiving DDP. The inclusion criterion for the study was that the families were currently receiving and had received at least one session of DDP with their child. The participants could be receiving DDP remotely (online), face-to-face or a hybrid of both delivery methods. Therapists involved in the RIGHT trial who deliver DDP were approached to aid in the identification of protentional participants. Email invitations to participate in the study were sent to families receiving DDP and participants recruited purposively until an adequate sample size with a variety of delivery

**Table 1. Parent and child demographics.**

| Parents Name | Childs Name | Child age (years) | Adopted/Long Term Fostered | Parent DDP Session (Online/Face-to-Face) | Parent and Child DDP Sessions (Online/Face-to-Face/Hybrid) |
|---|---|---|---|---|---|
| Jill (1) | James | 12 | Fostered | Online | Online |
| Hannah (2) | Megan | 13 | Adopted | Online | Online |
| Anne (3) | Daniel | 10 | Adopted | Online | Hybrid |
| Henry (4) | Jason | 16 | Adopted | Online | Hybrid |
| Susan (5) | Matthew | 11 | Adopted | Online | Face-to-face |
| Stephen and Greg (6) | Isaac | 6 | Adopted | Online | Face-to-face |

N.B. The names of the parents and children have been changed to protect anonymity

methods and child ages was reached. The recruitment period started on 25th January 2023 and ended on 18th March 2023.

## Data collection

A semi-structured topic guide was created to ensure that the interview was in line with Interpretive Phenomenological Analysis (IPA) [37, 39]. IPA allows researchers to gain insight into participants' lived experience [35]. As a result, the questions were open-ended, and did not lead the participants to elicit a specific response. A lead qualitative researcher along with two assistant researchers reviewed the topic guide and revised the questions to ensure relevant topics were included in line with the aims of the study. Questions covered family background, the child's experience of face-to-face or remote DDP, perceived effectiveness of DDP and expectations for future sessions. Participants were asked to highlight any struggles their child faced in relation to the delivery method, and what aspects of remote or face-to-face DDP their child enjoyed. A flowing dialogue between participant and researcher was created to gain rich information in line with the aims and methodological approach of the study.

## Procedure

All of the interviews were conducted via video call through Microsoft teams or Zoom. The interviews (n = 6) were between 37 minutes and 1 hour 44 minutes long (Mean = 1 hour and 20.83 minutes).

## Analysis

Interviews were recorded, transcribed verbatim, anonymised and stored securely. Analysis was carried out in accordance with IPA principles by two researchers [36]. Transcripts were read multiple times to enable familiarisation of the text, followed by interesting sections being highlighted and annotated. Emerging themes were identified from the notes, and connections made between them giving rise to a group of themes. This process was repeated for each transcript. All transcripts were then cross referenced for similar groups of themes and a title given to represent a group. Finally, the themes were drawn together into a table of superordinate themes and subthemes. The transcripts were analysed independently by the two researchers and identified themes discussed to clarify interpretations and ensure agreement between researchers.

### Research team and reflexivity

The research team included one post-doctoral researcher who led interviews and two under-graduate students who conducted all the analysis. All researchers and authors involved in the study are female. All researchers and authors were not involved in delivery of DDP session to the participants. It is recognised that researcher bias invariably affects analysis when using IPA. Researchers debriefed after each interview to voice and reflect on personal feelings which arose during the interviews. Recurrent group meeting were also held throughout the analysis process to discuss emerging ideas and how researcher's subjective experience influenced their interpretative framework.

## Results

4 superordinate themes and 12 subthemes were interpreted from the data, displayed in Table 2.

### Environment and child engagement

The physical environment where therapy sessions took place was an important factor influencing families' experience of DDP. Families were divided in their experience however, with some citing the home environment as advantageous and others expressing preference for being in a therapy room. Furthermore, the experience of these families illuminated a clear link between the environment and the child's engagement with DDP. This link was often very individual and unique to the child's circumstances and characteristics.

**Safety, comfort, and vulnerability.** Families receiving remote DDP found a sense of comfort and safety at home. Hannah felt that being in a private space enabled her child, Megan, to be vulnerable and approach difficult conversations with the therapist more easily:

> *"it's really quiet up there, she's in her own environment . . .her comfortable area, her bedroom is very much the way she likes it . . . I should imagine it helps massively, typically when you're talking about things that are tricky, anxiety, anger, you know."–Hannah (Interview 2)*

Hannah highlighted how Megan had a greater sense of self when in her own bedroom which makes it easier for her to navigate difficult emotions and past experiences during DDP. Having control over her therapy setting may have alleviated the constraints of potential power imbalances within the therapeutic relationship. When online, Hannah senses that Megan can better share power with the therapist which allows her to enter uncomfortable territory when she feels ready. For Jill, the importance of a familiar environment is also valued, however the individual meaning given to being at home is one that is more fundamentally related to her child's ability to engage with DDP at all. Contrasting their experience of remote DDP to previous experiences out with the home environment, Jill positions DDP as "working perfectly" for James, interpreted to mean that it is entirely suited to his unique needs:

**Table 2. Superordinate themes and sub-themes.**

| Environment and Child engagement | Travel | Non-verbal communication | Familiarity with Remote Interactions |
|---|---|---|---|
| Safety, comfort, and vulnerability | 'The driving back was decompression' | Observing and reacting to body language | 'New normal' |
| 'Being in the room' | Travelling is worth it | Importance of the therapist | 'I don't know how you'd do it online' |
| Role of the screen | 'Online makes it all possible' | | 'Digital native' |
| 'A very clear boundary' | | | |

*"So, leaving my side, going to therapy or being involved with the study, outside of the home environment would have been difficult for him . . .the virtual meetings are working perfectly, it's convenient."–Jill (Interview 1)*

By removing a sense of intimidation and uncertainty associated with face-to-face therapy, remote DDP has allowed James to receive support in a way that is on his own terms, in a place he feels safe and secure. Both Hannah and Jill's children have benefited most from DDP being conducted in the familiar home environment and enabling a sense of safety.

In contrast, families receiving face-to-face therapy described the importance of **'being in the room'** for the child's engagement within DDP. Stephen and Greg discuss how the physical environment contributed to their child's breakthrough moment during a DDP session:

*"Isaac did seem to really open-up and kind of started to talk about something that happened when he was a baby. . .I think it really was about being in the room at that point . . .there was a cake in the reception . . . and it was actually whilst he was eating that, that it happened . . . I think it was kind of a particular set of circumstances."–Stephen and Greg (Interview 6)*

For this family, being in the therapy centre cultivated unique conversations that the parents feel may not have happened without the nuances of the in-person environment. The therapist used the physical room including toys and cake to create a sense of warmth and comfort for Isaac, helping him to feel safe to share a difficult past experience. Susan also linked the environment to her child, Matthew's, engagement and progress in DDP. In this experience, however, there was a focus on the therapist's actions in the room:

*"Yeah, I think when he's in a room, when he went into the room with the therapist and they would be interacting, you know, drawing, that really sucked him in. . . he was sitting next to her [the therapist] on the floor talking to her."–Susan (Interview 5)*

Susan perceived that being in the same physical room allowed Matthew to naturally interact with the therapist, using the shared activity of play. In their first DDP session together as a family, Susan saw play and the positioning of the therapist next to Matthew as a vehicle by which her child formed a positive relationship with the therapist. This was a factor contributing to her preference for and view of face-to-face DDP being the superior form of delivery.

**Role of the screen.** During remote DDP, the environment comprises both the home and the computer screen. The screen itself took on various roles that meant different things for each participant–both experienced and hypothesised. For Hannah, the screen was perceived as a familiar medium of communication that her child has normalised, thereby aiding her ability to communicate:

*"I think, because young people of her age are so comfortable with screens, I think it's almost easier to say stuff and share stuff."–Hannah (Interview 2)*

Other parents saw the screen as a hindrance to engagement; for example, acting as a distraction:

*"I don't think he would have engaged as much.I think the screen is a distraction, especially with other faces, it's a big distraction, he's very easily distracted, so I don't think that would have worked very well."–Susan (Interview 5)*

Despite not having first-hand experience of conducting DDP online, Susan believed Matthew's impaired attention would impede his progress if their DDP was via remote delivery. Anne relayed a similar view, linking the screen to a lack of authenticity:

*"I think he would not have been able to have taken it as seriously, I think it would have been just going through the motions, just sort of attending and saying what he thought people wanted to hear. I don't think we would be where we are now, we wouldn't be as far as we are, I think it would have taken a lot longer."–Anne (Interview 3)*

Like Susan, Anne assumed that her child, Daniel, would not engage well with DDP through the screen, which would mean a much lengthier process in terms of any progress. She suggested that this would be due to a sense of reality being missing or perhaps because Daniel wouldn't be making a meaningful connection with the therapist. On the other hand, having experienced remote DDP, Jill described being surprised by her child, James's, capability to adapt to communicating via a screen:

*"His concentration level is very poor, but for some reason . . .the whole on-line learning and zoom calls and so-forth throughout the beginning of the pandemic worked so well for him surprisingly, because he was able to just sit there and concentrate and that was not him [pre-covid], I found it incredibly satisfying."–Jill (Interview 1).*

Like the other parents, Jill was concerned that her child would struggle to concentrate online. However, this assumption was dispelled after seeing James engage effectively in remote DDP. Finally, Henry pointed out a unique advantage of online DDP that is directly related to the role of the screen:

*"but one of the real positive things was on the screen, especially from the physical contact because it's one of Jason's things he doesn't necessarily like, especially if he's trying to isolate, he will put himself at the other side of the room . . . One of the things we realise. . .was having the screen automatically put him next to you."–Henry (Interview 4)*

Here, Henry described realising that the screen allowed him to have physical contact with his son which he previously hadn't experienced in face-to-face DDP. He saw this as an incredibly important stepping stone towards resolving Jason's attachment difficulties and in building the parent-child relationship.

*'A very clear boundary'* illustrates parents view of the benefit of separating home and therapy environments:

*"I think maybe there is something in the idea that we go to this special place to do this thing, and then we go away again. . .there is a very clear boundary which may well be a good thing."–Stephen and Greg (Interview 6)*

For Stephen and Greg, partitioning DDP from day-to-day home life was appealing as it kept stressful situations, such as upsetting emotional reactions from their child in response to topics discussed during DDP, out of their home environment. Furthermore, their child responded well going to a new environment and it formed part of his DDP experience; the parents' use of the words "special place" suggests they had created positive connotations of the therapy centre for Isaac. Generating a sense of excitement and curiosity surrounding DDP for Isaac may have improved his engagement within sessions. This may not be the same if DDP

was in the home environment hence this family was hesitant to adopt an online DDP approach. Indeed, Henry identified the difficulty his family experienced transitioning back into daily life during remote DDP:

> *"The trouble of being home is right at the end you're walking from your front room into the kitchen, and you haven't got any decompression."*–Henry (Interview 4)

After online sessions, Henry conveyed that the family carried their difficult emotions back into normal life as they did not have built in time or space to regulate and reflect at the end of sessions. The importance of being able to decompress via travel time is now further detailed in the next theme.

## Travel

Every family mentioned the impact of travelling to a therapy centre in their interviews. There was a divide in opinion, with some families seeing travelling as a means of releasing emotions after a session. Other families found long travel times to be unfeasible, and not having to travel to sessions took pressure away from the experience. Despite their long travel time to the sessions, some families believed that DDP was worth any inconvenience.

**'The driving back was decompression'.**   Despite Stephen and Greg's long travel time, they see the journey as part of the experience, and travelling helps them and their child regulate their emotions after a DDP session:

> *"It's just helpful to compartmentalise it, you know."*–Stephen and Greg (Interview 6)

This family can use their travel time to reflect on their emotions together and effectively transition between sessions and their home-life.Henry reflected that Jason will often use the journey time to sleep after a session, providing further separation between therapy and home:

> *"Definitely for Jason the driving back was decompression, as I said, quite often he would literally just go to sleep for the twenty minutes, half an hour."*–Henry (Interview 4)

As DDP is often very emotionally challenging, and traumatic events are discussed, having a period of time where they can slowly transition back to their home life after the session allows parents and children to co-regulate emotions and retain a sense of balance. Henry noticed that remote DDP, and the lack of journey time, resulted in a sudden transition between therapy and their home life:

> *"So, I think at home it's that. . .we didn't see it straightaway, but if we didn't engage with him [following the DDP session] we quite often saw more challenging behaviour after the remote sessions if we didn't manage the post-session element."*–Henry (Interview 4)

For Henry, being with Jason on the journey home allowed him to feel supported in understanding what happened in the session. It allowed for Henry to be present during the immediate time after a session, when Jason's behaviour can become challenging. For this family, being able to manage post-session emotions was just as important as emotions during the sessions. Often, travelling with the child can be an organic way to discuss feelings and thoughts, and manage potential negative reactions to difficult emerging emotions.

**Travelling is worth it.**   For Anne, the 45-minute journey each way, including emissions charges, is worth it:

*"In the beginning with the thought of the travel and getting there and it was the paramount thing . . . it was hard work to push to have to go. Seeing the relationship change. . .that tells me that it's worth it."–Anne (Interview 3)*

The progress that families make in therapy is perceived by them as a welcome step forward. This suggests that face-to-face therapy is perceived as being well worth the travel by some participants.

**'Online makes it all possible'.** In contrast, Stephen and Greg have a 90 minute trip each way to attend DDP sessions with their child via public transport, and they are at their limit:

*"In terms of travel, so if it was the choice between doing this with a two-hour journey, or doing it at home, I think practically we'd have to do it at home."–Stephen and Greg (Interview 6)*

This suggests that remote therapy could lighten the burden for these families and ensure families can still access DDP when physical constraints make it difficult for families to attend therapy centres. The potential for widened access of DDP to rural families is epitomised by Hannah's experience. Hannah lives on an island 140 miles from the therapy centre, so in-person sessions were difficult and stressful for her to attend. Hannah is thankful for remote DDP sessions, as she would do anything to help Megan, and having an online option made it possible for her to continue to access care:

*"I'd have travelled to outer Mongolia to try and get Megan the help and get us the support that we needed . . .we would have done it, but it would have added to the stress of what was already a very stressful situation."–Hannah (Interview 2)*

Hannah was in desperate need of care for Megan, in particular, she wanted help from DDP therapists to support her child's specific needs, but this was not available in her location. By allowing DDP to be accessible in a remote setting, Hannah no longer had to make a choice between stressful journeys or giving her child the specialist care that she needs.

## Non-verbal communication

Families highlighted the importance and usefulness of being able to observe and react to body language during DDP sessions. For that reason, loss or limited non-verbal communication perceived by many families was identified as a major drawback in online sessions. Some participants reflected on their own experience of communicating in online sessions, and others acknowledge the therapist's perspective during this time. The were contrasting experiences with some families pointing to the skills and experience of the therapist as factors contributing to their positive experience of non-verbal communication online. Even where there was a loss of connection felt, these skills allowed the screen to not be a barrier to non-verbal communication. Some families also found that non-verbal communication could still be captured with the right set-up of screens.

**Observing and reacting to body language.** Despite advocating for remote DDP as it fits well for her son's needs, Jill identified limited non-verbal communication as a downside:

*"You miss the personal connection, you know, the interaction. I think that's one of the main downsides of on-line, virtual meetings, it's not so easy to read people's body-language."–Jill (Interview 1)*

Here, Jill linked her feeling of not personally connecting with the therapist to her inability to see their body language. It may be that she struggled to feel in-sync and attuned with the therapist without being able to see their full body. Jill's experience alludes to non-verbal communication acting as an important factor towards enabling a strong therapeutic relationship. Greg and James shared similar views, but by taking on the perspective of the therapist:

*"I think Adam[DDP therapist] is also able to see how Isaac's reacting to things much more easily. I think also the difficulty of being on-line is that Isaac has to be in one place for it to work, whereas in the room Isaac can actually wander around a bit, play with this, look at that."–Stephen and Greg (Interview 6)*

Stephen and Greg placed value on body language as a tool for the therapist. They also expressed their assumption that Isaac would be expected to sit in front of the screen if online and believed this would negatively affect the dynamics of DDP. Their concern may be rooted in fear of losing the playful environment which is not only important for their child's enjoyment within DDP, but also impacts on PACE and therefore therapy processes. Whilst Stephen and Greg's views of online DDP are hypothetical because they have only had face-to-face DDP with their child, Anne's family have experienced both modes of delivery and she has similar concerns about body language. When comparing the two experiences, she highlighted body language as being a "massive difference":

*"for our children you are almost cutting off half the body, you are cutting off half of their communication. . .So, where you are cutting half that body language off for them, it's almost as if you're covering an ear or covering an eye. I think, it does put them at a bit of a disadvantage."–Anne (Interview 3)*

Anne went on to describe body language as a vehicle for children both in their ability to understand others better and to express themselves.Online DDP was seen as jeopardising this form of communication. Overall, Anne's experience highlights how perceived loss of body language may play a large role in parent's preference for face-to-face therapy. In contrast, Hannah felt that online DDP still captured body language due to screen set-up:

*"I think, as far as looking at Adam is concerned, and seeing how he's responding to what we're saying, that's absolutely fine, because we see him . . .the way we set screens up, we can see, most of him, and we would be the same as if we were there in person"–Hannah (Interview 2)*

Hannah and her daughter found that they were able to communicate both verbally and non-verbally with their therapist online. Overall, Hannah's family report that they have maintained the dyadic nature of DDP when online and have enjoyed their experience.

**Importance of the therapist.** Many families reported that the communication skills of therapist played an essential role in having success in online DDP:

*"I think it's down to the person [therapist], not just down to the child. It's the person, the way the person on that screen is interacting with a child, because they can only see their face, so the smile, the nods, the still making that sort of eye contact."–Jill (Interview 1)*

Having previously described a feeling of missed 'personal connection' when online, Jill emphasised the simple gestures a therapist can enact to convey listening, interest, and understanding. Jill felt these skills contributed to James's engagement and positive experience of

online DDP.Henry also discussed an example of how the therapist was able to counteract the loss of body language by verbally checking for it during online DDP:

*"I think that Adam was conscious of the fact that he couldn't observe the full body, so sometimes when he was facilitating the session. . . he would take a step or two back, and go over something and just check." - Henry (Interview 4)*

Henry elaborated on his experience of transitioning from face-to-face to online DDP; discussing the importance of establishing trust first:

*"we'd gone to that space and there was a degree of trust between me and Adam and Jason, so that you could have the harder conversations."–Henry (Interview 4)*

There is a sense of the unknown when moving to online DDP, however, Henry suggested that forming a strong therapeutic relationship by establishing trust in the face-to-face setting can enable families to continue their progress online with ease. Henry's family trusts their therapist, which allows them to tackle difficult topics. They also trust their therapist's professional skills to be able to facilitate these conversations and ensure body language is still being assessed during remote DDP. The combination of these factors allowed Henry's family to continue their positive experience of DDP after moving online.

### Familiarity with remote interactions

A clear relationship between previous experience of remote interactions and acceptability of remote DDP was interpreted from the data. This was particularly evident in parent's reports of older children who are said to be confident using technology and more accepting of online DDP.

**'New normal'.**   All families conducted their initial parent-only session remotely, and due to the COVID-19 restrictions of the time, some families had already adjusted to communicating online in various forms:

*"I think it's come after two years of everybody having quite a lot of practice at doing things remotely. . .so I think it was fine to do it that way."–Stephen and Greg (Interview 6)*

*"I'm usually fine, because we zoom all the time, it's something we've got used to isn't it really."–Hannah (Interview 2)*

Here, these families conveyed that their previous exposure to remote interactions positively influenced their experience during online DDP. Hannah linked this to online schooling experience for her daughter, Megan:

*"because of COVID. . . we had to do all that home-schooling, so she's quite comfortable, so the technology in itself isn't an issue."–Hannah (Interview 2)*

Whilst these families describe gaining confidence with technology during the period of the pandemic, one family shone light on how their child's familiarisation of screen was not new which is discussed in the following subtheme.

**'Digital native'.**   Henry's child Jason had been interacting with screens for longer than just the pandemic which helped him feel comfortable conducting their DDP sessions online:

*"Jason is a digital native, so he was very comfortable with that concept."–Henry (Interview 4)*

The phrase "digital native" epitomises how children that have grown up in a world of technology are also more open to participating in DDP remotely. Jason was able to translate his skills and knowledge from interacting in an online world to aid his engagement with therapy online. Whilst this family have no previous experience of DDP, they know how online interactions work and this has been a grounding factor in feeling confident to carry out online DDP.

**'I don't know how you'd do it online'.** Most of the families interviewed said they had little to no previous knowledge of DDP before starting the sessions. When this is coupled with a lack of digital experience, families are/were unsure of how DDP can work in a remote way:

> *"I don't know how you'd do it on-line . . .unless they are going to send us toys in the post, or, like he can use his own toys . . .I don't know what form it would take if we were to do it online."–Stephen and Greg (Interview 6)*

Stephen and Greg have been doing face-to-face sessions, and are still getting comfortable with the dyadic nature of DDP. As DDP utilises verbal and non-verbal cues within its concepts, such as a-r dialogue, they question how the interaction would work if online.

Susan and her child, Matthew, conduct all of their DDP sessions face to face, and Susan believes that Matthew has not acclimated to an online world:

> *"There is a barrier. . .I got used to talking openly with people, you know on the screen, but my son didn't. My son goes to primary school and, in the end, I ended up sending him in because I couldn't home-school him."–Susan (Interview 5)*

Again, the child's reported experience of online and home schooling was linked to how well the parent perceived that they would react to online DDP. Unlike Hannah's experience of Megan engaging in online schooling and therefore reacting well to online DDP, Susan believed that Matthew would not engage online because of his negative reaction to schooling at home. On the other hand, Hannah and her child had previous experience of online working, and now conduct all of their DDP sessions online:

> *"I think if we hadn't had the experience, then I think it would be much harder, possibly, but I think because we're so used to it, because of COVID and everything else that we don't really think about it."–Hannah (Interview 2)*

In Hannah's case, her previous experience of remote interactions allowed her to consider online DDP as an option for her, as there was a sense of familiarity to the concept. Previous knowledge of online environments has influenced the acceptability of online DDP for these families.

## Discussion

DDP is a family-based therapy aiming to treat maltreatment-related attachment difficulties in adopted children and long-term foster care. With the COVID-19 pandemic came the challenge of how to provide continuity of care for these families and therefore how best to translate practice from face-to-face to online. DDP is a unique relational and dyadic therapy which had not been delivered online prior to COVID-19 and no investigations into this mode of DDP delivery have been conducted. This study aimed to explore the experiences of families receiving remote DDP compared to face-to-face DDP. Importantly, this is the first study to explore online experiences of DDP. Interviews with parents brought to light four superordinate

themes central to this experience: *environment and child engagement*, *travel*, *non-verbal communication* and *familiarity with remote interactions*.

## Environment and child engagement

Parents discussed the role the environment played in their experience of DDP. For remote DDP families, being in the home environment was advantageous as it nurtured safety and comfort for the child. This is consistent with findings from other studies investigating online therapy experiences which highlight reduced feelings of threat and increased sense of safety for clients [40–42]. Safety in the therapeutic environment aids individuals to be less resistant and more open [43]. This is beneficial in promoting therapeutic processes in DDP, as feeling safe will allow the child to address traumatic events and moreover is an important feature within a securely attached relationship [44]. Whilst this is seen as an advantage of remote DDP, some families were concerned about boundaries as separating home life and therapy time was difficult for some families conducting DDP online. This concern is shared by therapists who also identified a blurring of boundaries when working online which importantly may impact on the perceived quality of the therapeutic relationship and further contribute to negative attitudes towards remote therapy [45]. Therapists with experience of online family therapy recommend families collaborate with their therapist to build new routines and rituals around remote therapy as this may assist in setting new boundaries and ensure easy transitioning between home life and DDP sessions [46, 47].

Many families linked the environment to their child's engagement with DDP, however parent's expressed mixed experiences. For the older children (aged 12 and 13 years), remote therapy overcame a major barrier to receiving support and enhanced their engagement. In adolescent populations, perceived stigma is recognised as an important barrier preventing engagement with services [48]. Online therapy removes stigma-related concerns surrounding going to a clinic, exemplifying how remote therapy can be leveraged to improve accessibility to DDP for these children [6]. Parents also described the screen as a positive barrier which aided their child to be open and have sensitive conversations. This experience is linked to the social disinhibition effect; where individuals feel greater freedom to be honest with the therapist online without fear of judgement [49]. For example, one study discovered that many gay and bisexual young men would opt for an online intervention method as opposed to face-to-face if given the choice as they felt unable to talk to a therapist in person about sensitive health issues [50]. This is useful in the context of DDP as children with attachment issues may struggle to trust the therapist which will impact on the effectiveness of therapy [44]. Conversely, for families with younger children, the screen was viewed as a major hindrance to engaging the child within the session. Parents' concerns over their child's engagement in online DDP is echoed by therapists who also expressed concern regarding maintaining a child's focus when translating practice to online [51, 52].

Having recognised the potential for distraction when working online with children, therapists and families also identified and developed various solutions to engage children effectively during the pandemic. Drawing together using the zoom whiteboard, using the chat function, watching videos together via screen share, playing online games and taking the therapist on a tour of the child's space and toys promoted children's engagement. Furthermore, these unique approaches to online communication also functioned as a means to convey emotions and facilitate difficult conversations [47, 52, 53]. Importantly, these creative uses of technology to facilitate interactions will not only enhance the child's attention but may further act as means to introduce the playfulness aspect of PACE into online DDP sessions. Additionally, some parents who expressed negative views of the screen, believing their child would engage less,

were thinking hypothetically having not experienced online DDP first hand. This outlook illuminated some parents' bias towards face-to-face therapy. Assessments of mental health professionals' attitudes revealed a general perception of online therapy as less effective; however they also recognised that negative attitudes can negatively influence the therapeutic relationship when working online [54]. Importantly, the researchers concluded that effectiveness is directly related to client characteristics such as patient age, computer literacy, and home environment. One parent did demonstrate a difference between assumptions and reality as her perception that screens are detrimental for concentration changed after seeing her child engage well with and benefit from DDP online, to her surprise. This family's experience highlights how parents and therapists cannot be certain about which children will engage with DDP in the online environment and which need an in-person approach. Much like how DDP adopts a curious stance, if parents can let go of rigid views surrounding the therapy environment and be open to trial and error, they can learn how environments impact their child's engagement in DDP sessions fist hand, reflect on the experience and use the information gained to adapt their therapy environment going forward.

## Travel

Every family interviewed in the study mentioned travel as either a positive or a negative aspect of DDP. For a number of families who were receiving face to face DDP, they relayed that travelling was a way to compartmentalise their feelings and decompress after a session. Being in a car or on public transport for 30 minutes to an hour or more allowed the children to talk about their feelings and the parents were able to guide them away from negative patterns of thinking. When clients are talking about stressors, triggering topics, or traumatic events, this can lead them to feel emotionally exhausted and can manifest as physical exhaustion. There is a positive link between childhood maltreatment and somatic symptom distress [55], therefore allowing oneself to transition from therapy to the personal realm in a controlled environment such as a car can be an effective way to maintain balance and reduce fatigue post-therapy. Sleep-related interventions could be used to complement the effects of psychotherapy in patients, and post-exposure naps have been found to reduce some indices of psychological stress in patients receiving exposure therapy for social anxiety [56, 57]. Travelling from a therapy session may give children a short window to reflect and re-consolidate their thoughts with few distractions. This is consistent with our findings and shows that travelling may help to build consistent post-session routines. Furthermore, parents reported that when they are near to their children in the car, they feel more able to help them with emotional regulation techniques. This was reportedly difficult to achieve if the child was able to go to their bedroom to be alone for example. An important facet of DDP is co-regulation [19, 31], and families have reported that the transition period between the therapy centre and the home can help foster regulation patterns for the children. Emotional regulation is a dynamic process that can be refined over time by exposure to difficult emotions and implementation of regulation strategies [58]. Families who identified the car journey as a tool to give children time and space to emotionally regulate have shown its usefulness to allow to practice of response-focused emotional regulation which overtime, during DDP sessions, can be transformed to a reappraisal process where the child can change how they think about an experience to dampen the negative emotional impact and promote positive behaviour. Families who undertake DDP remotely may benefit from leaving the house for 15–20 minutes after a session to emulate the act of travelling home which can provide an effective transition for the child and aid development of an emotional regulation framework.

Families who were receiving DDP face-to-face believed that DDP was providing such positive results that travelling a long time to the sessions was not as much of a problem as opposed to if they had to travel for another type of therapy. They believed that DDP was worth the travelling. This is consistent with previous studies on DDP and family therapy [33]. However, this alludes to the idea that DDP might work best in person, and that remote therapy might feel like a second option for many families. Previous findings have suggested that most people would not choose remote therapy over a face-to-face option but would opt for remote therapy if there was no face-to-face alternative [59, 60]. However, adolescents are less likely to drop out of therapy if they are given the choice of remote or face-to-face delivery [61]. Attrition rates for therapy are higher if the young person is randomly assigned a delivery method. One study found that in mental health interventions for children in state care, high levels of commitment to the therapy will enhance efficacy rates [62]. Therefore, by allowing the family to choose between travelling or staying at home for therapy, DDP can be tailored to individual needs and circumstances.

Families who received DDP remotely noted that online sessions enabled them to receive DDP when they otherwise would not be able to do so. This includes families who live in remote locations, and families with children who struggle with travelling. By advancing remote therapy and adapting to different family's needs, DDP can become accessible to more families from a range of backgrounds and locations. Remote healthcare can help attract people that would otherwise have been unable to engage in support. One study conducted in Finland showed that increased travel time was associated with fewer mental health visits [63]. Those living more than 30 minutes away from their nearest health centre had fewer visits compared to those living 15 minutes or less from the centre. This shows that long travel times can be a barrier for patients accessing care which was a concern raised by families in this study. Importantly, this study shows that remote DDP can reach families who would not be willing or would be unable to receive DDP in person as demonstrated by those families who experienced success in DDP sessions conducted at home from an area where the service is not accessible at a short distance.

### Non-verbal communication

Parents unanimously identified body language as a crucial factor in DDP sessions and all but one family felt their ability to observe and react to non-verbal cues was compromised online. This view aligns with various studies into teletherapy which identify lack of non-verbal cues as a negative aspect of remote therapy [9, 11]. In DDP, great emphasis is placed on non-verbal communication as it is essential for building an affectively attuned therapeutic relationship which encourages the child to feel at ease to explore their past [64]. In addition, conveying acceptance, curiosity and empathy, all aspects of PACE, can be done through facial expression and movement which families fear may be lost or more difficult for children to understand when online. In DDP, it is important for the child to perceive empathy and behaviour that demonstrate interest as this allows the experience of such qualities oneself. An additional layer to this is the child's ability to reflect and understand their own state of mind, described as 'mentalisation' by Fonagy, which is also achieved by the child observing a reflective stance through the parents' body language [65, 66]. The capacity to mentalize is an important skill to be able to emotionally regulate, understand other's feelings and develop a stable sense of identity which can contribute to enabling secure attachment. Non-verbal communication difficulties online may therefore negatively affect DDP to a greater extent than other family therapy services. On the other hand, parents described having a strong therapeutic relationship, therapists' skills and correct use of the technology as factors which can overcome or mitigate against

challenges with non-verbal communication. These observations are supported by research which emphasises therapists use family members names, ensure appropriate camera placement and engage in interactive ways to solicit feedback such a using toys and objects to help guide sessions [67]. Previous DDP qualitative research further echoes this by identifying that families believe the skills of the therapist is an important aspect of DDP [33]. One family's experience characterises this as they voiced trusting their therapist to continue making progress in their DDP journey after switching to online during the pandemic. At the heart of DDP is the therapist's use of an intersubjective stance to create trust, safety and help the child develop a new positive autobiographical narrative [44]. Therapists employ empathy and active listening to convey their intersubjective experience of the child which are skills this family points towards as enabling them to trust the therapist when moving to online DDP. Overall, this family's experience of establishing a strong therapeutic alliance and intersubjectivity in-person highlights how this can help with continuation of therapy processes online.

## Familiarity with remote interactions

Most of the families mentioned that DDP was different to any therapy that they had tried before. As a result, they were unsure of how DDP would work in a remote setting. This unfamiliarity is consistent with other studies on remote healthcare [14, 33], as for a lot of clients, online therapy is uncharted territory. With COVID-19 propelling online schooling and zoom meetings into the forefront of our daily lives [68], more people are becoming familiar with remote interactions. This 'new normal' is allowing parents and children to navigate remote therapy in ways they would not have previously. Previous knowledge of technology and experience in the digital world has a positive relationship with acceptance of remote therapy [69]. This is true of the families interviewed in this study, as familiarity with remote work was linked to family's acceptability of remote DDP. Likewise, families whose children had negative, little, or no previous experience of online interactions expressed greater ambivalence towards conducting DDP online. Previous experience of remote interaction, particularly positive experience, may therefore be a potential predictor for effective remote DDP. Correlations between increased confidence and positive attitudes towards online therapy were elicited in studies assessing therapist experiences during COVID-19, aligning with the experience of remote DDP families [14, 45]. However, families who experienced face-to-face therapy perceived this as being more beneficial to them and disclosed notions that remote DDP would not be as successful. In particular, one family who were fixated on physical elements such as toys and a cake as factors which aided their child to open up during a session could not conceive how that could be replicated online. This disinclination towards DDP appeared to be rooted in fear of it not working. Indeed, for DDP families there is more at risk as there is currently no therapy with a strong evidence base for helping adopted children with complex histories of maltreatment, and traditional interventions such as CBT and play therapy are often not effective for these cohorts. Whilst therapists may be willing to transition to online therapy and embrace the challenges, families fear how this will affect their child's care and ultimately their treatment outcome [45].

## Strengths, limitations, and implications for future research

This study adds to the growing research surrounding DDP and is the first study to qualitatively analyse the strengths and barriers to remote DDP. Instead of extrapolating study results from other therapies, this study has collected and analysed first-hand accounts from parents who have experienced remote and face-to-face DDP with their child. Semi-structured interviews allowed parents to discuss their experience in detail without constraints of a questionnaire or

survey and an adequate sample size was recruited as similarities and differences were elicited in analysis with IPA. Overall, this study echoes the findings of Casswell et al. and Gurney-Smith & Wingfield [33, 70], suggesting that DDP needs to continue to be flexible and accessible for clients who do not respond to traditional, behaviour focussed interventions.

The salient insight of this study is that family experience of DDP is unique and individual. From this, clinicians should consider how the different delivery methods can be used to inform the other. For example, how can the perceived decompression or psychological barrier between home and DDP via travel be translated in an alternative form into remote DDP or how can the comfort of having a safe and familiar environment be introduced into face-to-face DDP? Ultimately, by offering families both delivery methods and by families having the opportunity to experience both methods, advantages and disadvantages can be found, reflected on, and implemented into future sessions. The skills and approach of the therapist seems pivotal to all experiences, often mitigating against perceived drawbacks of either mode of delivery.

To deepen our understanding of the participant experience of DDP and to address the limitations of this study, future research should consider including perspectives from children and from more families with comprehensive experiences of both delivery methods. This will allow for fair comparison of experience and shed light on how the assumptions towards remote DDP therapy could potentially be overcome. Furthermore, understanding the child's perspective of delivery method preference, where possible, will elicit further recommendations for therapists by families themselves. As travel was shown to be an important aspect of the DDP process, future research should explore the effects of geographical restraints and public transport on families' experience of DDP. This would allow healthcare providers to widen accessibility for remote family therapy into middle-and low income countries. Overall, by continuing this research and expanding our understanding of DDP experience, families and therapists will be better able to work towards tailoring the delivery of DDP to the child's characteristics and needs.

A recent longitudinal survey has shown DDP to be associated with improved child and caregiver wellbeing [71], however, an important question that remains to be answered is if, when delivered remotely, DDP stands up to the core values set out by its founder. This study aimed to assess families' lived experience of DDP delivery methods via parent and carer views, and therefore the results from this study cannot be used to make claims about the fidelity of online DDP. Our results do show that families can have positive experiences with online DDP, and for some families, DDP works best when it is delivered online. Randomised Controlled Trials alongside further qualitative research should be undertaken to generate further evidence of remote DDP's efficacy and alignment to the core principles of DDP. Potential results from such studies will have important implications for the equitable distribution of family therapy. If remote DDP is found to be a compromise of face-to-face therapy then families in remote, rural, or low-income countries may still experience health inequality. However, if remote DDP is demonstrated to be efficacious then families who prefer or require this delivery method can be assured that they are receiving high quality care and strategies to expand accessibility to DDP can be evaluated.

## Conclusion

This study investigating families' experience of remote and face-to-face DDP via semi-structured interviews with parents and carers showed that the environment, the ability of the child to engage, the act of travelling to therapy, the role of body language and the family's previous experience of remote interactions were all important factors in the perceived success and quality of remote therapy or face-to-face therapy. Positive and negative aspects were found to be

associated with each theme indicating that what works for one family does not necessarily work for another. This study suggests that healthcare providers would benefit from giving families a choice between remote and face-to-face DDP and that there are learning points from parent/carer experiences of online delivery that can inform the face to-face approach, and vice versa. The introduction of remote DDP has broadened our opportunity for learning about DDP more generally and in ways that may otherwise not have been unearthed. By introducing comparison between modes, parents therefore reflect on aspects of face-to-face DDP delivery that they may not have considered before online modes were available (e.g. the role of travel in the DDP experience, the impact of the therapeutic physical context on the child and barriers to some children in engaging in 'in person' therapy). Not only does the introduction of online DDP offer alternatives to the challenges of face-to-face delivery, but also allows us to dissect in greater depth the role of the various ways in which face-to-face DDP operates to improve outcomes for families.

Therapists should therefore work collaboratively with families; assessing how the environment impacts the child's engagement levels, how travel influences the DDP experience, how parents perceive the impact of potential differences of body language online and how their previous experience of remote work may shape their preference for the DDP delivery method. Tackling these challenges will be individual and unique to each family so therapists will have to be creative and co-create new strategies with each family.

## Supporting information

**S1 File. Interview excerpts.**
(DOCX)

## Acknowledgments

Many thanks to Irene O'Neil who transcribed all interviews. Thanks to Karen Crawford for coordinating this project and leading ethical guidance. Thanks to Ben Gurney Smith for reviewing an early version of the paper and providing useful feedback.

## Author Contributions

**Conceptualization:** Helen Minnis.

**Data curation:** Monica Blair, Leigh Tweedlie, Fiona Turner.

**Formal analysis:** Monica Blair, Leigh Tweedlie.

**Investigation:** Monica Blair, Leigh Tweedlie, Fiona Turner.

**Methodology:** Fiona Turner.

**Supervision:** Helen Minnis, Fiona Turner.

**Visualization:** Fiona Turner.

**Writing – original draft:** Monica Blair, Leigh Tweedlie.

**Writing – review & editing:** Monica Blair, Leigh Tweedlie, Helen Minnis, Irene Cronin, Fiona Turner.

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
