## [Decision Letter · Decision Letter 0]

20 Nov 2023

PONE-D-23-22258Online therapy with families - what can families tell us about how to do this well? A qualitative study assessing families' experience of remote Dyadic Developmental Psychotherapy compared to face-to-face therapyPLOS ONE

Dear Dr. Blair,

Thank you for submitting your manuscript to PLOS ONE. After careful consideration, we feel that it has merit but does not fully meet PLOS ONE’s publication criteria as it currently stands. Therefore, we invite you to submit a revised version of the manuscript that addresses the points raised during the review process.

We look forward to receiving your revised manuscript.

Kind regards,

Zhengwei Huang

Academic Editor

PLOS ONE

Journal Requirements:

When submitting your revision, we need you to address these additional requirements. 1. Please ensure that your manuscript meets PLOS ONE's style requirements, including those for file naming. The PLOS ONE style templates can be found at https://journals.plos.org/plosone/s/file?id=wjVg/PLOSOne_formatting_sample_main_body.pdf and https://journals.plos.org/plosone/s/file?id=ba62/PLOSOne_formatting_sample_title_authors_affiliations.pdf 2. Note from Emily Chenette, Editor in Chief of PLOS ONE, and Iain Hrynaszkiewicz, Director of Open Research Solutions at PLOS: Did you know that depositing data in a repository is associated with up to a 25% citation advantage (https://doi.org/10.1371/journal.pone.0230416)? If you’ve not already done so, consider depositing your raw data in a repository to ensure your work is read, appreciated and cited by the largest possible audience. You’ll also earn an Accessible Data icon on your published paper if you deposit your data in any participating repository (https://plos.org/open-science/open-data/#accessible-data). 3. In your Data Availability statement, you have not specified where the minimal data set underlying the results described in your manuscript can be found. PLOS defines a study's minimal data set as the underlying data used to reach the conclusions drawn in the manuscript and any additional data required to replicate the reported study findings in their entirety. All PLOS journals require that the minimal data set be made fully available. For more information about our data policy, please see http://journals.plos.org/plosone/s/data-availability. Upon re-submitting your revised manuscript, please upload your study’s minimal underlying data set as either Supporting Information files or to a stable, public repository and include the relevant URLs, DOIs, or accession numbers within your revised cover letter. For a list of acceptable repositories, please see http://journals.plos.org/plosone/s/data-availability#loc-recommended-repositories. Any potentially identifying patient information must be fully anonymized. Important: If there are ethical or legal restrictions to sharing your data publicly, please explain these restrictions in detail. Please see our guidelines for more information on what we consider unacceptable restrictions to publicly sharing data: http://journals.plos.org/plosone/s/data-availability#loc-unacceptable-data-access-restrictions. Note that it is not acceptable for the authors to be the sole named individuals responsible for ensuring data access. We will update your Data Availability statement to reflect the information you provide in your cover letter. 4. Please include your full ethics statement in the ‘Methods’ section of your manuscript file. In your statement, please include the full name of the IRB or ethics committee who approved or waived your study, as well as whether or not you obtained informed written or verbal consent. If consent was waived for your study, please include this information in your statement as well. 

Additional Editor Comments:

Please revise your manuscript according to the reviewers' comments. Especially, please consider to apply the COREQ criteria mentioned by one of our reviewers.

Reviewers' comments:

Reviewer's Responses to Questions

**Comments to the Author**

1. Is the manuscript technically sound, and do the data support the conclusions?

Reviewer #1: Yes

Reviewer #2: Yes

2. Has the statistical analysis been performed appropriately and rigorously? 

Reviewer #1: N/A

Reviewer #2: Yes

3. Have the authors made all data underlying the findings in their manuscript fully available?

Reviewer #1: No

Reviewer #2: Yes

4. Is the manuscript presented in an intelligible fashion and written in standard English?

Reviewer #1: Yes

Reviewer #2: Yes

5. Review Comments to the Author

Reviewer #1: This is an interesting qualitative study exploring the experience of adoptive parents (I'll comment later on the title of the article) receiving Dyadic Developmental Therapy (DDP); this study is part of a broader clinical trial that aims to assess the clinical and cost effectiveness of DDP compared to services as usual. The link between this qualitative study and the quantitative evaluation is clear and well justified.

Here are a number of remarks and comments which I hope will improve the rigor of this manuscript

INTRODUCTION

p.4 : more details are needed regarding the ethical concerns emerging regarding the negative impact of “tele-mental-health” (p. 4, li 75-76)

The paragraph on the qualitative studies carried out among therapists on their experience of online therapy is really interesting

There is a lack of transition between the first major part of the introduction on the evaluation of online therapies, and the second part on the Dyadic Developmental Psychotherapy.

Typo p. 5, li 103: “Rooted in attachment theory; the primary goal of DDP” the semicolon must be replaced by a comma”.

The goals and major therapeutic targets of the DDP are well described on the child's side, but we don't really understand what is being worked on with the adoptive parents. Viewing Table 1, I understand that the DDP includes sessions for the child and sessions for the parents? This must be clarified.

METHODS:

I strongly recommend that authors follow the COREQ criteria (Tong et al. 2007); most of the information required by this checklist is already mentioned in the manuscript, but some is missing. This guide would provide greater precision and clarity.

Are DDP sessions protocolised? More precisely, is there an average number of sessions? This question is linked to an inclusion criterion: “families (…) who had received at least one session of DDP”. Justification for this inclusion criterion? And I’m not sure the parents can fully answer to a question concerning their experience of therapy with only one session.

Some aspects of the procedure are unclear: if I understand correctly, the 6 families contacted agreed to participate to the study? How were the participants in the RIGHT trial selected? In the same way, how were the participants of Adoptionplus selected?

Do the authors, as therapists, know the participants before the study? Were they involved, and in what capacity, in patients’care?

The choice of IPA is insufficiently justified; it is a very specific method which does not only allow access to the lived experience of individuals (a thematic analyze also allows this). Attention to the meaning-making processes is central, and the specific contribution of this method is not well understood here. I'll continue my comments on IPA in the results section.

RESULTS

See the new terminology that Smith and his colleagues have adopted: http://www.ipa.bbk.ac.uk/news/events

Some subthemes appear to be quite redundant. For instance, the subtheme “a very clear boundary” (p. 13) is very close to the superordinate theme of “travel”, especially with the subtheme “the driving back was decompression” (these two elements are by the way discussed together in the discussion section).

The part about car/public transportation journeys after the session and everything that goes on in them is really very interesting. This could be given greater prominence, particularly in the discussion section.

Unfortunately, this study can be criticized for its lack of conceptual and methodological precision, which is precisely what is being criticized in the Smith’s methods. More precisely: what exactly is a theme (superordinate or subtheme)? What is really phenomenological in the results? Where is the interpretative account? See Van Manen, M. (2017). But is it phenomenology? (Editorial). Qualitative Health Research, 27, 775–779. See also Van Manen, M. (2018). Rebuttal rejoinder: Present IPA for what it is—Interpretative psychological analysis. Qualitative health research, 28(12), 1959-1968.

I think that if the authors were to clarify a number of points concerning the recruitment procedure, other limitations would become apparent (selection bias, for example). All studies have biases, of course, it's just a question of being more precise and transparent.

Reading the results, it's clear that this is not about the family experience of DDP, but about parents’ experience.

DISCUSSION

The results are well discussed in the light of previous studies, but the discussion section does not provide any additional elements concerning the meaning-making processes regarding the therapy experience, which should nevertheless be at the heart of an IPA. I therefore suggest that the authors inject more elements regarding the processes at work in the therapeutic relationship, which they cite without detailing; for example, emotional regulation and the fact that this is a process that needs time to be worked on (link with the theme of the journey; see for instance Gross, J. J. (2001). Emotion regulation in adulthood: Timing is everything. Current directions in psychological science, 10(6), 214-219. Or: Cole, P. M., & Hollenstein, T. (Eds.). (2018). Emotion regulation: A matter of time. Routledge.). Similarly, the identification of therapist skills could be linked to empathy and the importance of intersubjectivity in child development (see Fonagy’s work).

In short, the discussion really needs to be given a more phenomenological consistency; otherwise the study, which is really of good quality, is "only" a thematic analysis (which is quite valid, but does not correspond to what is advertised).

I hope these remarks will help the authors in the very important field of evaluating the quality of our interventions!

Reviewer #2: Blair et al. reported on an interesting topic on dyadic developmental psychotherapy. The manuscript is well-structured and comprehensive, and the conclusion is solid. However, there are still some minor issues that should be addressed before acceptance.

1.The abstract is too long. It is suggested to cut the abstract and concentrate it to be more logical.

2.The language should be carefully polished. There are some inappropriate sentences and words, which should be carefully revised.

3.Please double-check the format of the references.

4.The discussion part should be more in-depth. The current version is too short.

6. PLOS authors have the option to publish the peer review history of their article (what does this mean?). If published, this will include your full peer review and any attached files.

Reviewer #1: **Yes: **Léonor FASSE

Reviewer #2: No

---

## [Author Response · Author response to Decision Letter 0]

5 Mar 2024

Dear PLOS ONE,

Thank you for taking the time to review our paper and provide feedback. The comments provided by the editor and reviewers were welcomed and extremely useful for us. We have worked hard to incorporate the suggested revisions into a new manuscript which we believe now has more clarity regarding the methodology and an increased depth of discussion.

We would like to address one specific comment from the editor here which regards our data availability:

Due to the nature of our study we collected data in the form of transcripts from interviews with participants. The interview transcripts contain personal and sensitive data. Data cannot be shared beyond individual quotations in the manuscript as participants did not give consent to the publication of interview transcripts in full. All relevant data can be found in the manuscript as direct quotations from participants which align with our study aim. This data can also be found in the supporting information file. 

All other responses relevant to reviewer comments are described in depth in the response to reviewers document.

Kind Regards,

Monica Blair

---

## [Decision Letter · Decision Letter 1]

19 Mar 2024

Online therapy with families - what can families tell us about how to do this well? A qualitative study assessing families' experience of remote Dyadic Developmental Psychotherapy compared to face-to-face therapy

PONE-D-23-22258R1

Dear Dr. Monica Blair,

We’re pleased to inform you that your manuscript has been judged scientifically suitable for publication and will be formally accepted for publication once it meets all outstanding technical requirements.

An invoice for payment will follow shortly after the formal acceptance. To ensure an efficient process, please log into Editorial Manager at Editorial Manager® , click the 'Update My Information' link at the top of the page, and double check that your user information is up-to-date. If you have any billing related questions, please contact our Author Billing department directly at authorbilling@plos.org.

Kind regards,

Zhengwei Huang

Academic Editor

PLOS ONE

Additional Editor Comments (optional):

After checking the revised manuscript and the new comments from the second round of review, I supposed that the current version had been well improved, and could be considered for publication in PLoS One.

Reviewers' comments:

Reviewer's Responses to Questions

**Comments to the Author**

1. If the authors have adequately addressed your comments raised in a previous round of review and you feel that this manuscript is now acceptable for publication, you may indicate that here to bypass the “Comments to the Author” section, enter your conflict of interest statement in the “Confidential to Editor” section, and submit your "Accept" recommendation.

Reviewer #2: All comments have been addressed

2. Is the manuscript technically sound, and do the data support the conclusions?

Reviewer #2: Yes

3. Has the statistical analysis been performed appropriately and rigorously? 

Reviewer #2: Yes

4. Have the authors made all data underlying the findings in their manuscript fully available?

Reviewer #2: Yes

5. Is the manuscript presented in an intelligible fashion and written in standard English?

Reviewer #2: Yes

6. Review Comments to the Author

Reviewer #2: (No Response)

7. PLOS authors have the option to publish the peer review history of their article (what does this mean?). If published, this will include your full peer review and any attached files.

Reviewer #2: No

---

## [Editor Report · Acceptance letter]

24 Mar 2024

PONE-D-23-22258R1 

PLOS ONE

Dear Dr. Blair, 

I'm pleased to inform you that your manuscript has been deemed suitable for publication in PLOS ONE. Congratulations! Your manuscript is now being handed over to our production team.

Kind regards, 

on behalf of

Dr. Zhengwei Huang 

Academic Editor

PLOS ONE